# Design and Characterization of a Novel Hapten and Preparation of Monoclonal Antibody for Detecting Atrazine

**DOI:** 10.3390/foods11121726

**Published:** 2022-06-13

**Authors:** Lingyuan Xu, A.M. Abd El-Aty, Jae-Han Shim, Jong-Bang Eun, Xingmei Lei, Jing Zhao, Xiuyuan Zhang, Xueyan Cui, Yongxin She, Fen Jin, Lufei Zheng, Jing Wang, Maojun Jin, Bruce D. Hammock

**Affiliations:** 1Institute of Quality Standard and Testing Technology for Agro-Products, Chinese Academy of Agricultural Sciences, Beijing 100081, China; xulingyuan163@163.com (L.X.); leixingmei163@163.com (X.L.); zj13048453218@163.com (J.Z.); zhangxiuyuan2017@163.com (X.Z.); cxy18763802353@163.com (X.C.); 0891syx@163.com (Y.S.); jinfenbj@163.com (F.J.); zhenglufei@caas.cn (L.Z.); w_jing2001@126.com (J.W.); 2Department of Pharmacology, Faculty of Veterinary Medicine, Cairo University, Giza 12211, Egypt; abdelaty44@hotmail.com; 3Department of Medical Pharmacology, Medical Faculty, Ataturk University, Erzurum 25240, Turkey; 4Natural Products Chemistry Laboratory, Biotechnology Research Institute, Chonnam National University, Yongbong-ro, Buk-gu, Gwangju 500-757, Korea; jhshim@jnu.ac.kr; 5Department of Food Science and Technology, Chonnam National University, Gwangju 500-757, Korea; jbeun@jnu.ac.kr; 6Department of Entomology & Nematology and the UC Davis Comprehensive Cancer Center, University of California, Davis, CA 95616, USA; bdhammock@ucdavis.edu

**Keywords:** atrazine, hapten, monoclonal antibody, immunoassay

## Abstract

This study provides the first design and synthetic protocol for preparing highly sensitive and specific atrazine (ATR) monoclonal antibodies (mAbs). In this work, a previously unreported hapten, 2-chloro-4-ethylamino-6-isopropylamino-1,3,5-triazine, was designed and synthesized, which maximally exposed the characteristic amino group ATR to an animal immune system to induce the expected antibody. The molecular weight of the ATR hapten was 259.69 Da, and its purity was 97.8%. The properties of the anti-ATR mAb were systematically characterized. One 9F5 mAb, which can detect ATR, was obtained with an IC_50_ value (the concentration of analyte that produced 50% inhibition of ATR) of 1.678 µg/L for ATR. The molecular weight for the purified 9F5 mAb was approximately 52 kDa for the heavy chain and 15 kDa for the light chain. The anti-ATR mAb prepared in this study was the IgG_1_ type. The working range of the standard curve (IC_20_ (the concentration of analyte that produced 20% inhibition of ATR)_—_IC_80_ (the concentration of analyte that produced 80% inhibition of ATR)) was 0.384 to 11.565 µg/L. The prepared anti-ATR mAb had high specificity, sensitivity, and affinity with low cross-reactivity. The prepared anti-ATR mAb could provide the core raw material for establishing an ATR immunoassay.

## 1. Introduction

Atrazine (2-chloro-4-diethylamino-6-isopropylamino-1,3,5-triazine, ATR) is an extensively used selective systemic triazine herbicide [1]. It resists environmental degradation owing to a slow photolysis and hydrolysis rate [2]. It disrupts the photosynthetic (energy-producing) process in broadleaf weeds and annual grasses [3]. ATR has become the most frequently detected pesticide in surface water and groundwater [4]. It has a long residence time and a stable structure. Therefore, it remains active for several years, causing environmental pollution [5]. Because of its low toxicity, it allows affected organisms to survive long-term, inducing subacute damage. Large amounts of pesticide residues thus remain in crops, water, and soil. This can cause severe damage to living organisms and disrupt the ecological balance [6]. ATR has carcinogenicity, teratogenicity, and mutagenicity. It affects the reproductive, endocrine, central nervous, and immune systems [7,8,9]. Therefore, it is imperative to establish a rapid and highly sensitive qualitative and quantitative detection technique for ATR residues.

Many countries have established residue limits for ATR. The European Food Safety Authority (EFSA) has assessed the maximum residue limit (MRL) of ATR in cereals to be 0.1 mg/kg. The European Union requires that the mass concentration of a single pesticide in drinking water should not exceed 0.1 μg/L and that the total mass concentration of all pesticides should not exceed 0.5 μg/L. In Australia, the MRL for ATR in corn, sugarcane, and sorghum is 0.1 mg/kg. The US EPA has set the MRL for ATR in fat, meat, and meat byproducts to 0.02 mg/kg and the MRL for ATR in drinking water and corn to 0.1 μg/kg and 0.25 mg/kg, respectively. In China, the MRL for ATR in drinking water (GB 5749-2006) and surface water (GB 3838-2002) is 2 and 3 μg/L, respectively.

The conventional detection methods used for ATR residue analysis include instrumental and immunoassay methods. Chromatographic analysis methods include gas chromatography (GC) [10], gas chromatography–mass spectrometry (GC/MS) [11], high-performance liquid chromatography (HPLC) [12], liquid chromatography–mass spectrometry (HPLC/MS) [13], and ultra high performance liquid chromatography (UHPLC) [14]. These methods are highly accurate and sensitive. However, they require complex sample pretreatment, expensive instrumentation, and cumbersome operation, which do not allow for extensive testing. Immunological assays have the advantages of simplicity, rapidity, and accuracy, meeting the requirements of modern rapid detection techniques. Immunoassays are based on the principle of the specific recognition of antigens and antibodies. Antibodies serve as the basis for immunological detection methods. The preparation of antibodies with high specificity, sensitivity, and affinity has become the most important part of improvements in detection techniques.

This study designed and synthesized an ATR hapten to retain part of its active structure. The purity, molecular weight, and structure were identified. The prepared ATR hapten was pure and structurally accurate. The ATR complete antigen was prepared using the active ester method. Mouse monoclonal antibodies against ATR were obtained by immunizing mice using immunogen, cell fusion, and hybridoma screening. The anti-ATR monoclonal antibody prepared in this study was the IgG_1_ type. The availability of monoclonal antibodies can lay the foundation for establishing immunoassays, enabling the rapid and high-throughput detection of ATR. The high specificity, sensitivity, and affinity of the murine monoclonal antibody against ATR could meet the requirements for the rapid, high sensitivity, and selectivity screening of ATR residues in agricultural products and environmental contamination in the field.

## 2. Materials and Reagents

Melamine, sodium bicarbonate, ethanol, and dichloromethane were procured from Beijing Chemical Factory Co. (Beijing, China) N,N-diisopropylethylamine (DIEA) and 3-aminobutyric acid were acquired from Beijing Evenhe Technology Co. (Beijing, China). The ATR standard was acquired from First Standard (Tianjin, China). N,N’-Dicyclohexylcarbodiimide (DMF) was supplied by Tianjin Seans Biochemical Technology Co. (Tianjin, China). 1-Ethyl-(3-dimethylaminopropyl) carboimide hydrochloride (EDC), N-hydroxysuccinimide (NHS), polyethylene glycol 2000(PEG-2000), cell freezing medium dimethyl sulfoxide (DMSO; serum-free), hypoxanthine, aminopterin, and thymidine (HAT), hypoxanthine and thymidine (HT) medium supplements, penicillin, streptomycin, L-glutamine, horse-radish-peroxidase-labeled goat anti-mouse IgG, complete and incomplete Freund’s adjuvant, TMB (3,30,5,50-tetramethyl benzidine) substrate solution, 1-(3-dimethylaminopropyl)-3-ethylcarbodiimide hydrochloride (EDC), bovine serum albumin (BSA), ovalbumin (OVA), 50% (*w*/*v*) polyethylene glycol solution, and dimethyl sulfoxide (DMSO) were provided by Sigma–Aldrich (St. Louis, MO, USA). Cell culture medium (Dulbecco’s modified Eagle’s medium; DMEM) and fetal bovine serum (FBS) were obtained from Gibco BRL (Paisley, Scotland). GIBCO^®^ Australian Premium FBS, GIBCO^®^ DMEM basic (1X) basal culture medium, penicillin solution, and L-glutamine solution were picked up from Thermo Fisher Scientific( Waltham, MA, USA). Analytical-grade sodium bicarbonate (NaHCO_3_), sodium carbonate (Na_2_CO_3_), sodium chloride (NaCl), sodium hydrogen phosphate dodecahydrate disodium hydrogen phosphate dodecahydrate (Na_2_HPO_4_·12H_2_O), potassium dihydrogen phosphate (KH_2_PO_4_), citric acid monohydrate (C_6_H_10_O_8_), Tween-20, hydrogen peroxide (H_2_O_2_), and gelatin were purchased from Sinopharm Chemical Reagent Co. (Beijing, China). The mouse Sp2/0-Ag14 myeloma cell line was purchased from the Cell Resource Center of Peking Union Medical College (Beijing, China). Triazine standards (ATR, terbuthylazine, simetryn, propytryn, terbuthylazine, simazine; 99% purity) were secured from First Standard (Tianjin, China). Horseradish peroxidase (HRP) was purchased from Jackson Immunoresearch Laboratories Co. (West Grove, PA, USA). Cell culture plates (6-well™, 24-well™, and 96-well™) and a 96-MicroWell™ transparent plate were secured from Costar (Corning, USA). Cell pipettes (1, 5, and 10 mL) were purchased from Thermo Fisher Scientific (Thermo, Vantaa, Finland). Syringes (1 and 5 mL) were purchased from Shandong Zhu Pharmaceutical Co. (Shandong, China). The glassware used in the experiments was strictly cleaned and sterilized before use. The glassware, tips, and reagents were sterilized by autoclaving. The autoclave was set at 120 °C, and items were sterilized for 20 min. After sterilization, they were placed in an oven, allowed to dry, and then used.

## 3. Methods

### 3.1. Synthesis of ATR Hapten

The synthesis route of the hapten is shown in Figure 1a. Briefly, the ATR hapten was first obtained by synthesizing product SM_1_ and then reacting SM_1_ with 3-aminobutyric acid to obtain the desired hapten. The intermediate SM_1_ was synthesized by dissolving melamine (0.92 g, 5.0 mmol) in acetonitrile (50 mL). The solution was cooled to 0 °C in an ice-water bath. The cooled solution was mixed with an aqueous solution of ethylamine (0.99 mL, 5.0 mmol) and diisopropylethylamine (DIEA) (2.65 mL, 15.0 mmol). The reaction mixture was stirred at 0 °C for 4 h and then dried by spin evaporation. The crude product obtained was separated by silica gel column chromatography (petroleum ether:ethyl acetate = 5:1) to obtain product SM_1_ (4,6-dichloro-N-ethyl-1,3,5-triazine-2-amine), 780 mg (81%). SM_2_ was synthesized by dissolving SM_1_ (935 mg, 5 mmol) in 20 mL ethanol. Then, SM_2_ (515 mg, 5 mmol) and DIPEA (1.94 g, 15 mmol) were mixed. The reaction solution was heated to 85 °C for 3 h. The complete reaction of SM_1_ was monitored using thin-layer chromatography (TLC). The solution was then cooled to room temperature at 25 ℃ and spin-dried. Saturated aqueous sodium bicarbonate solution (20 mL) was added to the spin-dried sample and stirred for 20 min. The reaction was completed, and the products were extracted twice with 20 mL of dichloromethane. The aqueous phase was adjusted with 1 M HCl (pH = 4). The white precipitate obtained from the reaction was filtered. After drying, 850 mg of ATR hapten was obtained. The recovery rate from this reaction was 62.5%. Finally, the hapten was characterized for purity, molecular weight, and structure using high-performance liquid chromatography (HPLC), high-resolution mass spectrometry (HRMS), nuclear magnetic resonance hydrogen spectroscopy (^1^H NMR), and carbon spectroscopy (^13^C NMR).

### 3.2. Preparation of Immunogen and Coating Antigen of ATR

A small-molecule hapten is only antigenic but not immunogenic. The immunogenicity of a small molecule hapten is achieved when it is linked to a large molecule (usually a protein). BSA has stable physical and chemical properties and many free amino groups and is cheap and easy to obtain. It still maintains great solubility under different pH values and ionic strengths and contains some organic solvents. It is often used as the carrier of immunogens. OVA can be used as a carrier protein for antibody screening and immunoassays because of its weak immunogenicity compared with other proteins. In this study, BSA and OVA were coupled with ATR hapten to prepare immunogen and coating antigen. The link between the hapten and carrier protein mainly occurs by coupling carboxyl and amino groups and other active wave groups. To obtain an ATR complete antigen with immunogenicity, the carboxyl group of the hapten was activated using the activating ester method and coupled to the carrier protein. The structural formulae for the activation of the hapten and the protein coupling reaction are shown in Figure 1b. The molar ratio of the hapten to the immunogen was 60:1, and the coating antigen was 50:1. The specific protocol was to add 3.15 mg (0.012 mmol) 3-(4-chloro-6-(ethylamino)-1,3,5-triazine-2-ylamino) butanoic acid, 2.79 mg (0.024 mmol) NHS, and 4.60 mg (0.024 mmol) EDC in 0.5 mL DMF. The solution was stirred magnetically overnight (10 h) in a refrigerator at 4 °C. After the reaction was completed, an activation solution containing the ATR hapten was obtained. This activation solution can be directly used for subsequent coupling with carrier proteins. After that, 20 mg of BSA (or 10 mg of OVA) was dissolved in 0.01 M PBS buffer (pH = 7.4, 10 mg/mL). Activated hapten was added to the BSA and OVA solutions drop by drop. The reactions were conducted under magnetic stirring at room temperature for 4 h. The resulting reaction solution was then dialyzed six times with 4 L of 0.01 mol/L PBS (pH 7.4). The dialysis removed unreacted hapten or other small molecules. After dialysis was completed, the antigen was dispensed at 1 mg/mL, snap-frozen in liquid nitrogen, and stored at −20 °C.

Immunogenicity was characterized by matrix-assisted laser desorption/ionization time-of-flight mass spectrometry (MALDI-TOF-MS). MALDI-TOF-MS measured the conjugation ratio of the coating antigen and immunogen. This method has the advantages of rapid and straightforward sample preparation and high sensitivity. The matrix auxiliary solution used for the measurement was configured as follows: the mixture (acetonitrile:water = 70:30) contained 0.001% trifluoroacetic acid and erucic acid (15 mg/mL). The wavelength of the Yag excitation light source was 355 nm. The *m*/*z* acquisition range was between 10 and 100 kDa. Each sample was mixed with the matrix auxiliary solution. After mixing, spot sample 1 was placed on the sample target of 1 μL. After natural drying, the sample was placed under a mass spectrometer for Yag laser scanning. The conjugation ratio was calculated as follows:Conjugation ratio = (*M*_p_ − *M*_std_)/*M*_h_.
where “*M*_p_” represents the ATR antigen conjugates, “*M*_std_” represents the BSA/OVA standard, and “*M*_h_” represents the ATR hapten.

## 4. Production and Characteristics of Anti-ATR mAb

### 4.1. Buffers and Solutions

(1) Coating buffer solution (CBS, pH 9.6, 0.05 mol/L): 2.93 g NaHCO_3_ and 1.5 g Na_2_CO_3_ were weighed and dissolved with Milli-Q water, and the volume was adjusted to 1 L. (2) Phosphate buffer solution (PBS, pH 7.4, 0.01 mol/L): 0.2 g KH_2_PO_4_, 2.96 g Na_2_HPO_4_•12H_2_O, and 8.0 g NaCl were weighed and dissolved in Milli-Q water, and the volume was adjusted to 1 L. (3) Washing buffer solution (PBST): PBS containing 0.1% Tween-20. (4) Sample dilution buffer solution (PBSTG): PBST containing 0.1% gelatin. (5) TMB chromogenic solution: three solutions are currently used and configured. The formula of each enzyme label plate was as follows: 11 mL matrix buffer (46.04 g potassium dihydrogen citrate hydrate and 0.10 g potassium sorbate dissolved to 1 L and stored at room temperature), 200 µL TMB stock solution (375 mg, 3,3,5,5′-tetramethylbenzidine dissolved in 30 mL dimethyl sulfoxide) and 101 µL 1% H_2_O_2_ (1 mL 30% H_2_O_2_ added to 29 mL Milli-Q water). (6) Termination solution (1 mol/L HCl): 44 mL of 98% hydrochloric acid was measured, 440 mL of deionized water was slowly added along the beaker wall, and the mixture was stirred while adding.

### 4.2. Immunization

Animal experiments were approved by the Experimental Animal Welfare and Ethical Committee of Institute of Quality Standards and Testing Technology for Agro-Products, Chinese Academy of Agricultural Sciences (IQSTAP-2021-05). Animal experiments were conducted in strict accordance with Chinese laws and guidelines. Six female Balb/c mice (7 weeks old) were immunized with immunogen (1 mL ATR-BSA (1 mg/mL, molar ratio of 19:1) + 1 mL Freund’s adjuvant (complete/incomplete)). The immunogen was mixed with Freund’s complete adjuvant for the initial immunization. Subsequent immunizations were performed by fully emulsifying the immunogen with an equal volume of Freund’s incomplete adjuvant. The immunization strategy for the mice is shown in Table 1. At 3–5 d after the third injection, the immunized mice were eye-bled, and the sera were tested for anti-ATR antibody titer and ATR recognition properties using ic-ELISA. The ic-ELISA protocol, buffers, and solutions were similar to those described previously [15]. The reaction was terminated by adding 50 μL of 1 M HCl per well. Optical density (OD) values at 450 nm were measured with an Infinite M200 PRO microplate reader (TECAN, Männedorf, Switzerland). The specificity of the developed anti-ATR serum was assessed using ic-ELISA. The inhibition rate was calculated according to the following equation:*IR* (%) = (1 − *B/B*_0_) × 100%
where “*IR*” represents the inhibition rate, B represents the OD_450nm_ value of the inhibition well, and *B*_0_ represents the OD_450nm_ value of the control wells.

### 4.3. Ic-ELISA Procedure

Ic-ELISA was applied to screen the serum of mice with the best performance fusion, detect the positive cell supernatant, and establish the standard curve. The operation steps are as follows: (1) Add the coated antigen to the microplate, and incubate each well with 100 µL at 37 ℃ for 30 min. Then, wash the plate with PBST 3 times. (2) Dilute the standard sample and antibody with PBSTG to the required concentration, add standard samples of different concentrations and 50 µL 9F5mAb in the microplate, and incubate at 37 ℃ for 30 min; wash the plate with PBST 3 times. (3) Add 100 µL IgG HRP diluted with PBSTG to each well; wash the plate with PBST 3 times. (4)Add 100 µL chromogenic solution to each well, avoid light at 25 ℃ for 15 min, and then add 50 µL hydrochloric acid solution to each well to terminate the reaction. The absorbance value was measured at OD450_nm_. In the whole process, 200 µL PBST was added to each well when washing the enzyme label plate.

### 4.4. Production of Anti-ATR mAb

#### 4.4.1. Preparation of Sp2/0 and Feeder Cells

One week before cell fusion, mice of the same strain used for culture and immunization were resuscitated and expanded. The specific operation steps of Sp2/0 (obtained from National Infrastructure of Cell Line Resource, Beijing, China) for myeloma cells are as follows: quickly transfer the cells from the liquid nitrogen tank to the 37 ℃ water bath until the solid is completely melted, and confirm that the cap of the cryopreservation tube has been tightened to prevent the cap from submerging into the water to prevent ice crystals from damaging the cell membrane. Preheat the DMEM culture solution to 37 ℃ in advance, slowly transfer the melted cell suspension to a 15 mL centrifuge tube containing 10–12 mL DMEM, blow it and mix it evenly, centrifuge at 1000 rpm/min for 10 min, and discard the supernatant. The cells were rung and dispersed at the bottom of the centrifuge tube on the super clean table. Then, 5 mL of 20% complete medium was slowly added to the cells to avoid too fast dropping acceleration and too large a change in cell osmotic pressure, resulting in cell wall rupture. After the resuspended cells were blown with a 5 mL pipette, they were transferred to a six-well plate with 2.5 mL per well and placed in a constant temperature incubator at 37 ℃ and 5% CO_2_. When the cells grew to 80% of the area at the bottom of the six-well plate, that is, in the logarithmic growth stage, they continued to be subcultured with 20% complete culture medium. The culture was expanded for two to three generations, and bone marrow cells with good growth state, uniform size, and good shading were selected for the subsequent fusion experiment of splenocytes and tumor cells.

A healthy female mouse over 10 weeks old was selected, the eyeball was removed, bleeding to death occurred, and the mouse was soaked in alcohol for 5 min. Fix the mouse with a needle on the anatomical table, with the abdomen upward, cut the epidermis with scissors, clamp the peritoneum with sterile tweezers, inject liquid with a disposable needle tube, gently squeeze the mouse’s abdomen with tweezers or shake its legs, draw out the liquid with a needle tube and repeat several times. The liquid was added to the extracted liquid, the supernatant was centrifuged and discarded, and the cells were resuspended in culture medium and placed in 96-well culture plates at 100 μL per well.

#### 4.4.2. Cell Fusion and Screening

The spleen cells collected from the mice were fused with the SP2/0 cell line using PEG-2000 at a ratio of 10:1 spleen:myeloma cells. The serum titer reached 6.4 × 10^4^ at an OD450 nm value higher than 1.0. This indicated good immunogenicity. The mice with the best inhibitory effect were selected to enhance immunity. Intraperitoneal injections of 0.05 mg immunogen (2 mg/mL, 0.025 mL) were conducted for the fusion. Three days after the last immunization booster, mouse spleen cells were separated and fused with PEG-2000 pretreated Sp2/0 myeloma cells to prepare hybridomas according to previously described procedures [16,17,18]. The cell culture medium formulations are listed in Table 2. The plates were incubated at 37 °C in a CO_2_ incubator (5% CO_2_ in air). The selective growth of the hybrid cells occurred in DMEM supplemented with 2% HAT. The fused cells were cultured in 2% HAT medium for 7 day. Several hybrid cells were screened seven days after fusion by testing the supernatant using ic-ELISA to determine the binding ability. Positive hybridomas were cloned by limiting dilution, and clones were further selected by ic-ELISA [19,20,21]. The anti-ATR mAb clone, designated 9F5, which had a high antibody titer and good sensitivity in the culture supernatant, was expanded in mice to produce mAb in ascites. The expanded cultured monoclonal cell line was injected into the abdominal cavity of mice treated with paraffin 1 week in advance to prepare the ascites antibody. Ic-ELISA was used to detect the efficacy and inhibition rate of mouse ascites. The antibodies were purified by ammonium sulfate precipitation [22,23]. The best antigen–antibody concentration was selected through checkerboard ic-ELISA and then established with the best combination. The standard curve was established to determine the sensitivity and detection range of the method. The purified antibody was dissolved in 0.01 M PBS and placed into a dialysis bag. It was dialyzed with 0.01 M PBS four times and PB two times, and the dialysate was changed every four hours. The dialyzed antibody protein was packed in a 1.5 mL centrifuge tube, frozen in liquid nitrogen, and vacuum lyophilized. The 9F5 mAb dissolved the lyophilized antibody powder with 0.01 M PBS and stored at −20 °C. Then, 50% glycerol was added and prepared at 1 mg/mL. This was used to detect the properties of monoclonal antibodies, including titer, specificity, and type. The assay cross-reactivity with ATR 9F5 mAb was determined using ic-ELISA. OriginPro 8.5 (OriginLab Corporation, Northampton, MA, USA) for Windows was used for data analysis. The OD_450nm_ values were plotted against the analyte concentration on a logarithmic scale, and the generated sigmoidal curve was mathematically fitted to a four-parameter logistic equation.

## 5. Results and Discussion

### 5.1. Identification Results of Hapten, Immunogen, and Coating Antigen of ATR

#### 5.1.1. Identification of Atrazine Hapten

The identification results of the ATR hapten are shown in Appendix A. The theoretical precise molecular weight [M + ^+^H] of the ATR hapten C_9_H_14_C_l_N_5_O_2_ was 259.69. According to the HPLC and HRMS results, the purity of the ATR hapten was 97.8%, as revealed by the peak area ratio. The peak positions indicated that the correct ATR hapten molecule was synthesized.

^1^H NMR (400 M*Hz*, DMSO-d6) δ12.16 (s, 1H), 7.86–7.61 (m, 2H), 4.27 (h, J = 6.8 *Hz*, 1H), 3.24 (dt, J = 14.1, 6.9 *Hz*, 2H), 2.60–2.51 (m, 0H), 2.32 (dt, J = 15.8, 6.4 *Hz*, 1H), 1.18–1.02 (m, 6H).

^13^C NMR (101 M*Hz*, DMSO) δ 172.37, 167.53, 165.07, 164.81, 164.54, 43.39, 43.12, 40.27, 40.12, 39.91, 39.70, 39.49, 39.28, 39.08, 38.87, 35.01, 34.84, 20.26, 20.00, 19.81, 14.59, 14.36, 14.23.

#### 5.1.2. Identification of Immunogen and Coating Antigen of ATR

The hapten and BSA feed ratios were 40:1, 50:1, and 60:1, respectively. The hapten and OVA feed ratios were 30:1, 40:1, and 50:1, respectively. The single charge ion peaks of BSA and OVA standards were 67132.709 and 44587.269, respectively. The molecular weight of ATR was 259.69. The average number of ATR hapten molecules coupled on each BSA and OVA by the active ester method can be calculated from the formula. The MALDI-TOF-MS results of the immunogen and coated antigen are shown in Appendix A. The coupling results of coated antigen and immunogen are listed in Table 3. According to the data, the immunogen selected by immunized mice was 60:1. The coated antigen ratio of ic-ELISA was 50:1.

### 5.2. Characterization and Cross-Reactivity of Anti-ATR mAb

#### 5.2.1. Characterization of Anti-ATR 9F5 mAb

The serum potency and inhibition rate of ATR mice are shown in Appendix A. The serum titer and inhibition rate of the fusion mice are shown in Appendix A. The titer and inhibition rate of mouse ascites before purification are shown in Appendix A. The optimal antibody and coating antigen concentrations for the inhibition of 100 ng/mL ATR were determined using the ic-ELISA method. The determined optimal concentrations of the antigens and antibodies are shown in Appendix A. Both the coated antigen and antibody were diluted to 2 × 10^3^, 4 × 10^3^, 8 × 10^3^, 1.6 × 10^4^, 3.2 × 10^4^, 6.4 × 10^4^, 1.28 × 10^5^, and 2.56 × 10^5^. They were used in subsequent experiments. The ATR standard was diluted to 100, 50, 25, 12.5, 6.25, 3.125, 1.5625, and 0 ng/mL in 0.01 M PBS for use in subsequent experiments. IC_50_ values from standard curves characterized the sensitivity of ic-ELISA under optimized conditions. The curve of ic-ELISA is shown in Figure 2a. Each value was the average of three independent repetitions. There was a good linear relationship between the inhibition rate of the antibody and the concentration of ATR. The IC_50_ of the anti-ATR 9F5 mAb was 1.678 µg/L, and the working range of the standard curve (IC_20_–IC_80_) was 0.384 to 11.565 µg/L.

According to the manufacturer’s instructions, the molecular weight of the antibody was determined using an SDS–PAGE kit (Solarbio^®^, Beijing, China). The prepared separation and concentrated gel concentrations were 10% and 5%, respectively. The purified 9F5 monoclonal antibody showed two bands with molecular weights of approximately 52 and 15 kDa, that is, the molecular weights of the heavy and light chains of the mAb. Other miscellaneous protein bands were smaller with a higher antibody purity, indicating that these could be used to establish an ATR immunoassay method. The SDS–PAGE results are shown in Figure 2b. The immunoglobulin isotype was determined with a mouse antibody isotyping kit (Sigma, St. Louis, MO, USA). Three replicate groups were used in the experiment. According to Table 4, the type of 9F5 ATR monoclonal antibody is IgG_1_.

#### 5.2.2. Cross-Reactivities of Anti-ATR mAb

The structure and cross-reaction rate of seven triazine herbicides are shown in Table 5. Although the cross-reaction rate between ATR and propazine was 41.50%, the cross-reactivities with other structural analogs were low. The existence of cross-reactions with other triazine herbicides is unavoidable because the chemical structures of triazine herbicides are very similar. This shows that the prepared 9F5 anti-ATR mAb had good specificity for ATR.

## 6. Conclusions

In this study, a novel 2-chloro-4-ethylamino-6-isopropylamino-1,3,5-triazine (ATR hapten) was designed and synthesized. The ic-ELISA method was initially developed, however, the condition parameters were not optimized. Furthermore, the preliminarily developed ic-ELISA was not applied for the determination of actual samples. It should be noted that the prepared anti-ATR mAb can act as a raw material for establishing an ATR immunoassay.

The hapten retained the chlorine substituent at R_1_, highlighting the antigenic determinant. Modifying the isopropylamine substituent at R_3_ and adding the connecting arm and active groups to the isopropylamine substituent can effectively reduce the cross-reaction rate of the antibody to compounds containing isopropylamine substituents to improve the specific recognition of triazine pesticides by the atrazine antibody.

The molecular weight of the ATR hapten was 259.69 Da, and its purity was 97.8%. The characteristics of the anti-ATR mAb were systematically investigated. The mAb 9F5, which can specifically detect ATR, was obtained, and its IC_50_ value was 1.678 µg/L. The anti-ATZ mAb prepared in this study is IgG_1_. The working range of the standard curve (IC_20_–IC_80_) was 0.384 to 11.565 µg/L. The molecular weights of the heavy and light chains of the purified 9F5 mAb were approximately 52 and 15 kDa, respectively. The prepared anti-ATR monoclonal antibody had high specificity, sensitivity, and affinity and low cross-reactivity.

The general structural formula of triazine herbicides is shown in Figure 3. The molecular formula of triazine herbicides that cross-react with mAb 9F5 is shown in Table 6. The previously reported atrazine hapten selectively introduces a carbon linkage arm and an active group (-CH_2_)nCOOH) at the R_1_ or R_2_ position. In this study, the cross-reaction with five other triazine herbicides with similar structures was determined. It can be seen that the antibody has a high cross-reactivity with propazine and ametryn, which are 41.50% and 18.86%, respectively. The reason for this is that their structures are very similar. The two compounds have an isopropylamine substituent at the R_3_ position. The designed hapten structure and cross-reaction rate are compared with previous reports, as shown in Table 7 [24,25,26,27]. The atrazine hapten prepared in this paper also has an ethylamine substituent at the R_3_ position. Therefore, the ethylamine substituent at the R_2_ position is a very important antigenic determinant of hapten. Some compounds without isopropylamine substituents and with methylthio or ethoxyl groups instead of Cl substituents at the R_1_ position showed low cross-reaction rates. In summary, we can conclude that the isopropylamine substituent at R_3_, Cl substituent at R_1_, and ethylamine at R_2_ also play a key role in the antigen–antibody recognition reaction.

## Figures and Tables

**Figure 1 foods-11-01726-f001:**
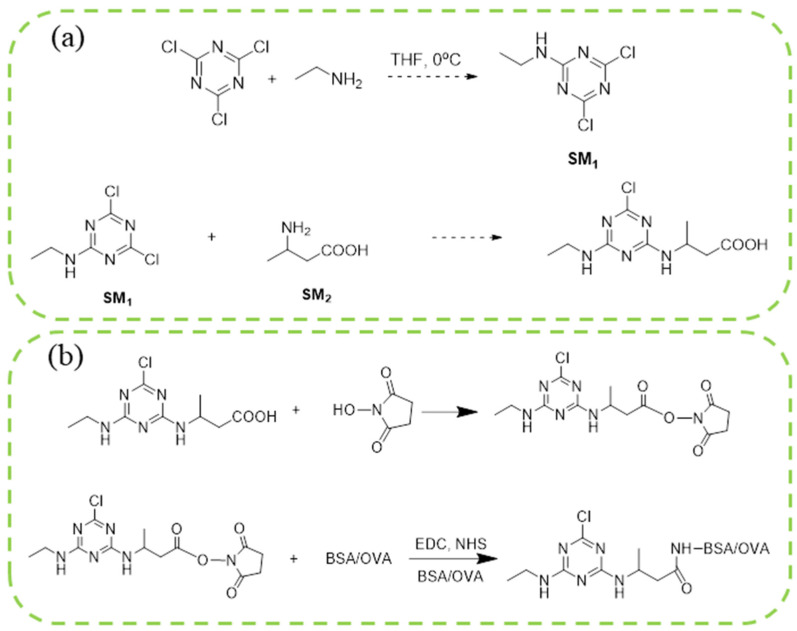
Synthetic routes of hapten (**a**) and immunogen and coating antigen (**b**) of ATR.

**Figure 2 foods-11-01726-f002:**
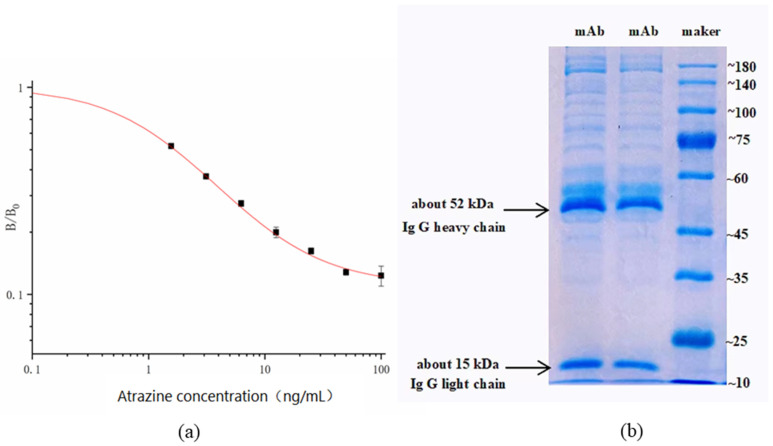
Ic-ELISA curve of the 9F5 mAb (**a**) and the SDS–PAGE results (**b**).

**Figure 3 foods-11-01726-f003:**
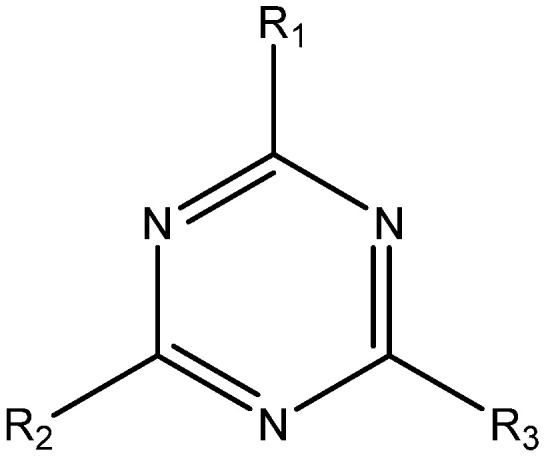
General structural formula of triazine herbicides.

**Table 1 foods-11-01726-t001:** Immunization strategy in mice.

Number of Immunizations	Immunization Cycle (Days)	Total Volume of Immunization (mL)	Immunization Dose (mg)
First	1	0.2 ^a^	0.1
Second	15	0.2 ^b^	0.1
Third	29	0.2 ^b^	0.1
Fourth	43	0.1	0.05

* ^a^—complete Freund’s adjuvant, ^b^—incomplete Freund’s adjuvant.

**Table 2 foods-11-01726-t002:** Cell culture medium formulations.

Name of Culture Medium	Formulation Volume Ratio (%)					
	DMEM Basal Culture Solution	L-Glutamine Solution	Penicillin Solution	50 × HAT Recovery Solution	DMSO	Fetal Bovine Serum
Complete culture solution	78	1	1	−	−	20
HAT complete Culture solution	76	1	1	2	−	20
Cell lyophilization solution	69	1	1	−	9	20

*−: no.

**Table 3 foods-11-01726-t003:** Identification of the immunogen and coated antigen of ATR.

**(A) Identification results for the immunogen.**				
**Hapten to Carrier Protein Dosing Ratio**	**Molecular Weight of Hapten-BSA (Da)**	**Mass Change** **∆m (Da)**	**Hapten to Carrier Protein Ratio**	**Hapten Number/BSA**
40:1	71,131.224	3998.515	14.46	14
50:1	71,413.430	4280.721	16.61	16
60:1	72,205.234	5072.525	19.05	19
**(B) Identification results of coated antigen.**				
**Hapten and Carrier Protein Dosing Ratio**	**Molecular Weight of Hapten-OVA (Da)**	**Mass Change** **∆m (Da)**	**Hapten to Carrier Protein Ratio**	**Hapten Number/OVA**
30:1	46,093.340	1415.697	5.45	5
40:1	46,113.244	1473.982	5.68	5
50:1	46,347.978	1727.179	6.65	6

**Table 4 foods-11-01726-t004:** ATR 9F5 mAb type.

Antibody Typing	Number of Times	IgG_2a_	IgG_2b_	IgG_3_	IgM	IgA	IgG_1_
OD_450nm_ value	1	0.9694	0.9774	0.9715	0.9704	0.9639	1.4213
	2	1.014	1.0106	0.9461	0.9825	0.984	1.4076
	3	0.9891	0.9609	0.9363	1.0191	1.0386	1.4222

**Table 5 foods-11-01726-t005:** Structures of triazine herbicides and their cross-reactivity rates.

Analytes	Chemical Structure	IC_50_ (ng/mL)	Cross-Reactivity
Atrazine	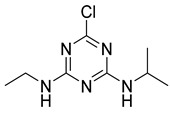	1.678	100.00%
Propazine	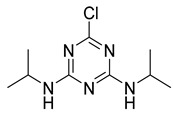	4.04	41.50%
Atrazine hapten	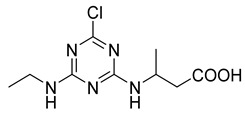	5.47	30.69%
Ametryn	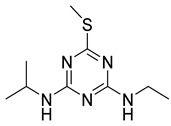	8.90	18.86%
Terbuthylazine	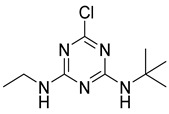	12.37	13.57%
Simazine	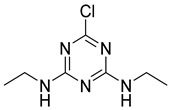	12.27	13.68%
Prometryn	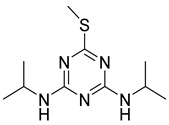	32.41	5.18%
Simetryn	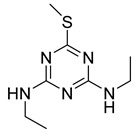	37.93	4.42%
Terbumeton	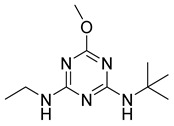	82.81	2.03%

Note: The concentrations of triazine pesticides were 100, 50, 25, 12.5, 6.25, 3.125, 1.5625, and 0 ng/mL.

**Table 6 foods-11-01726-t006:** Molecular formula of atrazine and triazine herbicides with cross-reaction.

Name	R_1_	R_2_	R_3_
Atrazine	Cl	NHCH_2_CH_3_	NHCH(CH_3_)_2_
Atrazine hapten	Cl	NHCH_2_CH_3_	NHCH(CH_3_)_2_COOH
Prometryne	SCH_3_	NHCH(CH_3_)_2_	NHCH(CH_3_)_2_
Economicpoison	OCH_3_	NHCH_2_CH_3_	NHC(CH_3_)_3_
Simazine	Cl	NHCH_2_CH_3_	NHCH_2_CH_3_
Terbuthylazine	SCH_3_	NHCH_2_CH_3_	NHC(CH_3_)_3_
Simetryn	SCH_3_	NHCH_2_CH_3_	NHCH_2_CH_3_

**Table 7 foods-11-01726-t007:** Comparison with previously reported atrazine hapten structures and cross-reaction rates.

Hapten Name	Hapten Structure	CR(%)	Reference
4-chloro-6- (isopropyl amino)-l,3,5-triazine-2-(6-ami- nohexanecarboxylic acid)	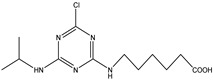	propazine (136%) cyanazine (28%) terbuthylazine (26%)	[24]
		AT-1-M8, AT-2-M1, and AT-3-M15 simazine (53-78%) AT-1-M3 MAb simazine (23%)	[25]
2-mercaptopropionic acid-4-ethylamino-6-isopropylamino-1,3,5-triazine	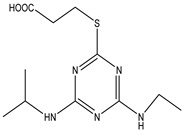	Simazine (19%) Melamine (0.043%) Chlorpyrifos (0.00012%) Monocrotophos (7.29 × 10^−7^%) Parathion (0.0079%)	[26]
4-((4-(isopropylamino)-6-methyl-1,3,5-triazin-2-yl)amino)butanoic acid	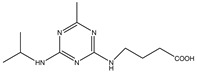	Simazine (2.5%) Propazine (90%) terbutylazine (5.0%) trietazine (11%) desmetryne (0.5%) ametryne (5.0%) atratone (6.4%) hydroxyatrazine (2.1%) hydroxysimazine (0.3%) hydroxypropazine (8.2%) hydroxyterbutylazine (5.6%) hydroxydesmetryne (0.1%)	[27]
2-chloro-4-ethylamino-6-isopropylamino-1,3,5-triazine	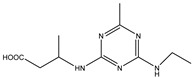	Propazine (41.50%) Atrazine hapten (30.69%) Ametryn (18.86%) Terbuthylazine (13.57%) Simazine (13.68%) Prometryn (5.18%) Simetryn (4.42%) Terbumeton (2.03%)	In this work

## Data Availability

The study did not report any data.

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
