# Peer review of "Design and Characterization of a Novel Hapten and Preparation of Monoclonal Antibody for Detecting Atrazine"

_foods, 2022, doi:10.3390/foods11121726_

Round 1

Reviewer 1 Report

In this manuscript, the authors have synthesized a new atrazine hapten and used it for the production and isolation of a new monoclonal antibody for detecting atrazine. Then, the isolated antibody was used for the construction of an indirect competitive-ELISA (ic-ELISA) method for the determination of atrazine. However, the manuscript needs a lot of improvement, and the supplied manuscript file lacks figures captions and supplementary; thus, I could not judge the manuscript correctly with the provided text only. I recommend major revision for the manuscript considering the following comments.

 1-      The authors should discuss the rationale and novelty of their designed atrazine hapten. Also, they should clarify how the designed hapten maximizes the exposure of the active group amino group of atrazine and highly induce the expected antibody. Moreover, they should compare their approach for designing the atrazine hapten with previously reported atrazine hapten.

2-      How the authors calculated the purity of the produced hapten.

3-      In table 5, it is clear that the produced antibody has high cross-reactivity with many pesticides which is in contrast to previously reported antibodies (International Immunopharmacology, 40, 2016, 480-486, 10.1016/j.intimp.2016.10.003). The authors need to comment on that.

4-      The authors should validate the developed ic-ELISA method in terms of LOD, LOQ, accuracy, and intraday and inter-day precision.

5-      The authors should test the performance of their method for the analysis of atrazine in real samples such as crops, environmental water, or soil samples.  

6-      The authors should clearly discuss the advantage of their produced monoclonal antibody, generated by their newly designed atrazine hapten, with previously isolated antibodies for atrazine.

7-      The authors should compare the performance of their developed ic-ELISA method with previously reported methods for the determination of atrazine in terms of sensitivity, selectivity, linear range, tested matrices, etc.

Reviewer 2 Report

The authors design and synthesis a novel hapten and develop of monoclonal antibody for atrazine. The novel antibody is with high affinity and sensitivity. The IC50 value is 1.678 µg/L and could meet the demand of MRL requirements of all countries. It is quite interesting and meaningful to the field of immunoassay. However, some revisions still need be addressed.

1. Line 54: Please revise “are” to “is”, and kindly revise the following text with this format.

2. Line 55: The “byproducts” should be “by-products”.

3. There should be a blank character between letters and numbers. Please check them all, such as Line 200, Line 332, and Line 295.

4. Line 267: Revise “myeloma” to “for myeloma”.

5. Line 281: “20% complete culture medium” should be revised to “a 20% complete culture medium”.

6. Line 290: Please kindly correct “Liquid” to “The liquid”.

7. There are some typos, grammatical errors, and missing space. Please check and revise the text.

Reviewer 3 Report

This work titled with “Design and characterization of a novel hapten and preparation

of monoclonal antibody for detecting atrazine”, is interesting describing the preparation of mAb from ATR hapten. The following comments:

1.      Line 26, ATZ should be replaced with ATR.

2.      The caption of the figures should be added.

3.      The authors are encouraged to present Figure 2 and 3 as the supporting materials.

4.      Please check the full manuscript throughout and make sure that all the font size should be uniform in the corresponding paragraph.

5.      What’s “DIPEA” on Line 147? “egg mass” on Line 174? “pbstg” Line 257/259?

6.      According to the context on Line 170, Ova was used as the carrier protein, however, it was replaced by BSA on Line 180. Please address it clearly.

7.      The context between Line 186 and 193 is encouraged to describe the hapten only with BSA, and then preparation of hapten with OVA followed with simply sentences.

8.      There is a wrong “)” on Line 235

9.      Statistical analysis and Ethical statement are not present.

10.  Native English-speaker is encouraged to revise the language grammar.

Round 2

Reviewer 1 Report

The authors have addressed some of the comments, however, other important comments were not addressed. For instance, the authors said that they could neither perform validation for the developed ic-ELISA method nor apply it to real samples. They said that their study is focusing on the synthesis of hapten and the characteristics of the antibody and currently no further experiment could be performed as Universities/ Labs are locked due to COVID-19.

Consequently, I think a limitation section should be added to the manuscript, in which the above authors' statements should be mentioned.

Author Response

This manuscript is a resubmission of an earlier submission. The following is a list of the peer review reports and author responses from that submission.